# Mechanical properties research of unconsolidated hydrate-bearing sediments under the effect of clay minerals

**Yuanwei Sun**[1,2], **Yuanfang Cheng**[1]*, **Cui Li**[2], **Liqiang Wang**[2], **Xiaodong Dai**[2], **Chuanliang Yan**[1]

**1** School of Petroleum Engineering, China University of Petroleum (East China), Qingdao, China,
**2** College of Petroleum Engineering, Shandong Institute of Petroleum and Chemical Technology, Dongying, China

* 2015004@sdipct.edu.cn

## Abstract

The shallow hydrate reservoir in the Shenhu Sea area is mainly composed of clayey silt. Clay mineral has an important impact on the mechanical properties, and the hydrate decomposition aggravates this impact. Therefore, the composition and geological conditions of shallow clay hydrate-bearing sediment in Shenhu sea area are fully considered, hydrate-bearing sediment samples with similar physical properties are synthesized in situ. Then, indoor triaxial mechanical experiments are carried out, and the effect of clay minerals on the mechanical property is analyzed. The results show that the clay content and clay type have an important impact on the mechanical properties of unconsolidated hydrate-bearing sediment. With the increase of clay content, the strain hardening characteristics are prominent, the yield stage is longer, and the plasticity is enhanced. Hydrate-bearing sediment with different clay content shows similar mechanical laws under the influence of hydrate saturation and effective confining pressure. The peak strength, elastic modulus and Poisson's ratio all show a downward trend, but the peak strength and elastic modulus change more obviously. The peak strength changes linearly with hydrate saturation, while nonlinearly with effective confining pressure, especially 0–3 MPa. This is the comprehesive result of clay particle's movement and fragmentation, clay hydration and expansion, affecting hydrate formation and sediment cementation. When the content ratio of montmorillonite/illite decreases, the peak strength and elastic modulus show an increasing trend. Because the frictional resistance and connection strength of illite crystal layer are larger with bigger particle size, weaker hydration and thinner water film. The research can provide reference for drilling and production engineering of natural gas hydrate (NGH) reservoir in the Shenhu sea area.

## 1. Introduction

As a new clean energy in the 21st century, NGH is characterized by abundant reserves and high energy density, which organic carbon is twice the total carbon of existing oil, natural gas and coal [1–3]. The exploration and development will help optimize the energy structure and reduce carbon emissions.

**Data availability statement:** All relevant data are within the paper.

**Funding:** This work had been financially supported by the Dongying Science Development Fund (DJ2023001), the National Natural Science Foundation Project of China (51974353, 51991362, 52104014), the Natural Science Foundation of Shandong Province (ZR2019ZD14), the CNPC Major Science and Technology Project (ZD2019–184–003) and the Project Establishment and Construction Team of Young and Innovative Talents Introduction and Education Plan of Colleges and Universities in Shandong Province.

**Competing interests:** The authors have declared that no competing interests exist.

The Shenhu sea area is in the middle of the northern continental slope of the South China Sea. The shallow hydrate reservoir is mainly clay silty sand, with a water depth of $1230 \sim 1245m$ and a seafloor temperature of less than 4 °C. The average ground temperature is 14.37 °C, and the formation pressure exceeds 10 MPa. The hydrate is distributed $183 \sim 225m$ below the mudline, and filled in the unconsolidated sediment in a dispersed form [4,5]. The sediment is mainly composed of terrigenous clastic mineral, clay mineral and little biocarbonate with a lower median pore radius of less than 1.5μm. The clastic mineral is mainly silty sand with particle size of 4-63μm [6]. The clay mineral is mainly illite and montmorillonite, accounting for more than 65%. The hydrate saturation is 16% ~ 45% and the porosity is 0.2–0.45 [6–10]. The hydrate decomposition leads to the change of formation pressure. The water rock interaction and seepage process aggravate the change of formation microstructure, resulting in complicated mechanical properties and deformation characteristics [9–20]. Therefore, the mineral composition, particle size distribution and physical properties of hydrate-bearing sediment have an important impact on the mechanical properties.

At present, researches have been carried out on the physical and mechanical properties of hydrate formation [21–27]. Indoor triaxial experiment is a commonly used research method. The experiment on the undisturbed samples of submarine NGH is carried out, and the result is compared with that of artificial hydrate samples and reconstituted hydrate samples [28,29]. It is believed that the particle size and cementation state will affect mechanical properties. The characteristics of natural hydrate-bearing samples, artificial hydrate-bearing samples and ice bearing samples are discussed, showing that the hydrate increases the shear strength and longitudinal wave velocity, but the longitudinal wave velocity of fine sediments is smaller than that of coarse sediments [30–32]. On this basis, the impact of filling types is further observed [33]. Then, the experimental conditions are analyzed, and the hydrate formation temperature has less effect on the mechanical properties, while the temperature, back pressure, effective confining pressure and hydrate saturation during triaxial experiments have significant effect [34–36]. The thermal, electrical, acoustic and mechanical properties are systematically studied, showing that the thermal conductivity is related to the phase state, saturation and spatial distribution, and the peak strength increases with the increase of hydrate saturation and confining pressure [37–39]. Now, the resonant column experiment is used to test the dynamic mechanical properties, and the dynamic elastic modulus and damping structure change law under confining pressure are explored [40,41].

During the NGH exploitation, the hydrate decomposition caused by temperature, pressure and phase equilibrium would weaken the formation strength, and influence the mechanical properties and constitutive relationship [41,42]. The strength and deformation characteristics in permafrost regions under different exploitation conditions are studied, showing that when the load is greater than the sediment strength after decomposition, thermal injection would lead to reservoir damage [43,44]. Kaolin is used to simulate submarine hydrate bearing sediments, showing that with the increase of confining pressure, the peak strength first increases, then remains flat, and finally decreases [45]. It is similar to the nature of frozen soil hydrate [46], because the effective confining pressure causes the pressure melting of ice crystals in the sample. The stress-strain curve of strongly cemented siltstone reservoir shows a typical rock stress-strain relationship under low effective confining pressure, which is different from the less obvious peak strength and strain hardening characteristics under high effective confining pressure [47].

In conclusion, researches on the mechanical properties of NGH have been carried out, but fail to consider NGH decomposition, clay content and clay type. Therefore, fully considering the geological and mechanical characteristics of NGH reservoir in the Shenhu sea area, hydrate samples with different clay content and type are synthesized in situ, triaxial experiments are carried out. Then the mechanical parameters and change laws under different clay content, clay type, effective confining pressure and hydrate saturation are obtained. The research is helpful to clarify the

mechanical characteristics and deformation characteristics of NGH reservoir, and provides basis and reference for the design and optimization of drilling and production engineering.

## 2. Experimental equipment and methods

### 2.1. Experimental equipment and process

The experimental device mainly includes the NGH in-situ synthesis system, low-temperature rock mechanics triaxial experiment system, core preparation device and physical property measurement instrument. The flow is shown in Fig 1.

The M190C microcomputer is used to control the low temperature cold storage for environmental constant temperature with a range of −25 °C to 50 °C and an accuracy of 0.1 °C. The TAW-100 microcomputer is adopted to control low temperature of triaxial testing machine. The axial pressure range is 0–100 KN, with an accuracy of 0.1 KN, the confining pressure range is 0–100 MPa, with an accuracy of 0.1 MPa, and the pore pressure range is 0–80 MPa, with an accuracy of 0.1 MPa. The axial deformation range of the sensor is 0–10mm, the radial deformation range is 0-6mm, and the measurement accuracy is 0.25% FS. The axial load is driven by ball screw, which avoids poor hydraulic oil fluidity at low temperature. The autoclave uses high quality carbon steel forgings. The chip deformation sensor is improved by using special low temperature viscose, suitable for low temperature and high pressure. The methane purity is 99.9%. The equipment is shown in Fig 2.

### 2.2. Synthesis conditions of hydrate sample

The gas hydrate phase equilibrium model [48] of porous medium is adopted, and the hydrate formation conditions and rate are comprehensively considered. The temperature is set at 2 °C. The model is shown in equation (1).

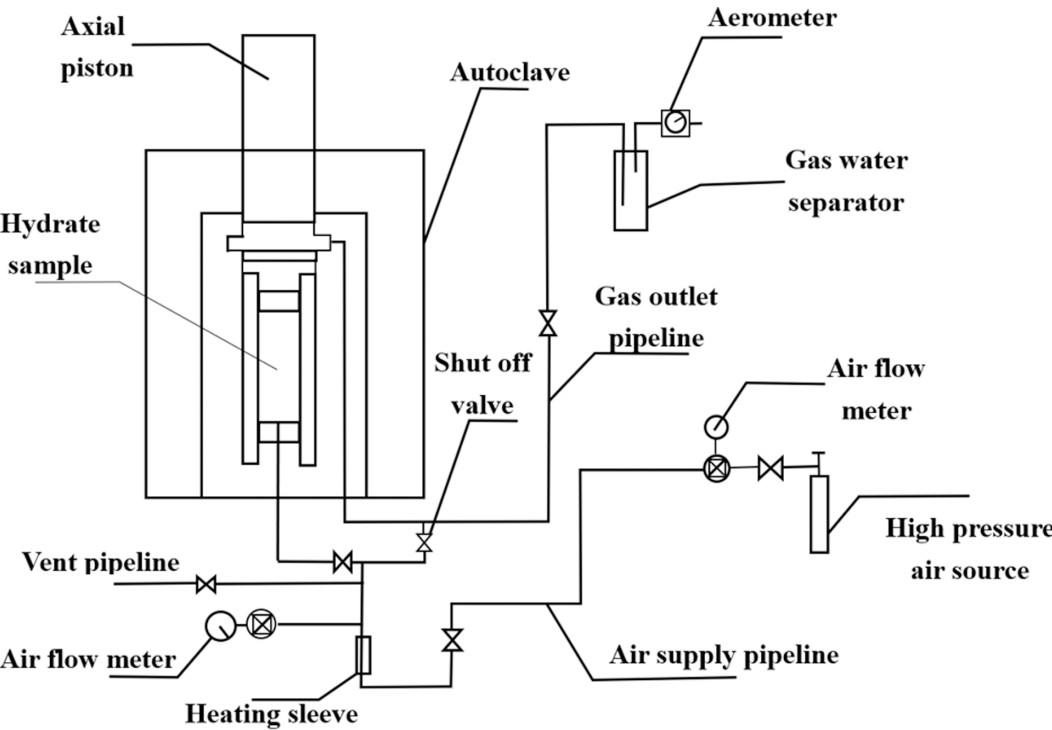

**Fig 1. Schematic for NGH in-situ synthesis system.**

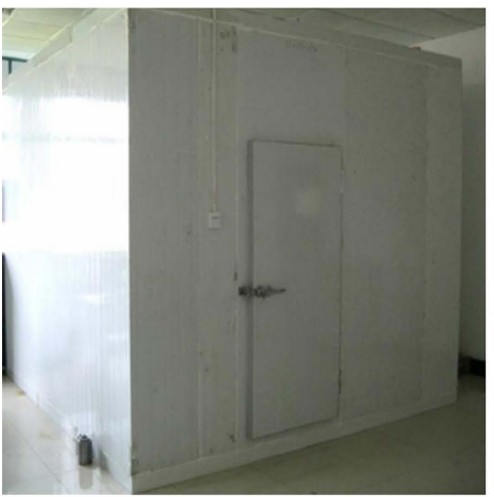
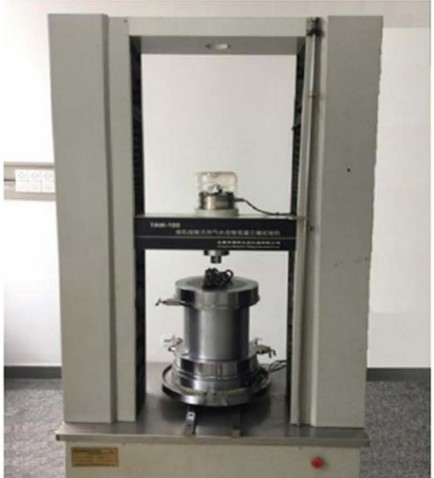
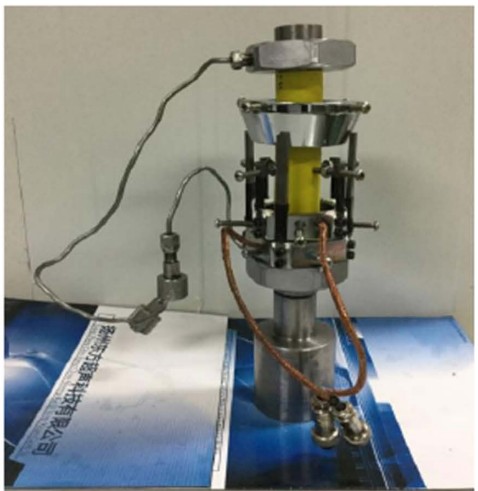

(A) cold storage          (B) triaxial testing machine          (C) core assembly equipment

**Fig 2. Experimental equipment.**

$$P = 8.13 \times 10^{-13} e^{0.1052T} \tag{1}$$

where, P is the pressure of NGH system, MPa; T is the temperature of NGH system, K.

The chemical equation for NGH formation is shown in equation (2).

$$CH_4 + nH_2O \Leftrightarrow CH_4 \cdot nH_2O \tag{2}$$

where, n is the reaction coefficient, 5.75.

The quality of $H_2O$ required to be added to the hydrate sample can be calculated by equation (3).

$$m_{H_2O} = \frac{207}{239} V \phi S_h \rho_h \tag{3}$$

where, $m_{H_2O}$ is the quality of water required, g; V is the total volume of hydrate sample, cm³; $\phi$ is the porosity, dimensionless; $S_h$ is the saturation, dimensionless; $\rho_h$ is the density of hydrate sample, g/cm³.

The "immersion+dropping" composite method is used to add water to ensure that the sample density error does not exceed 0.01g. While the amount of hydrate formation is verified by measuring the $CH_4$ volume after the experiment. The hydrate saturation is obtained comprehensively. The calculation method is as shown in equation (4).

$$\rho_t = \frac{m_{H_2O} + m_{core}}{V} \tag{4}$$

where, $\rho_t$ is the theoretical target density of hydrate sample, g/cm³; $m_{core}$ is the dry sample quality, g.

## 2.3. Hydrate sample test process

(1) Prepare quartz sand and clay, the particle size distribution is shown in Fig 3A and B respectively. A thermoplastic tube is placed inside the mold firstly. Then evenly mix the

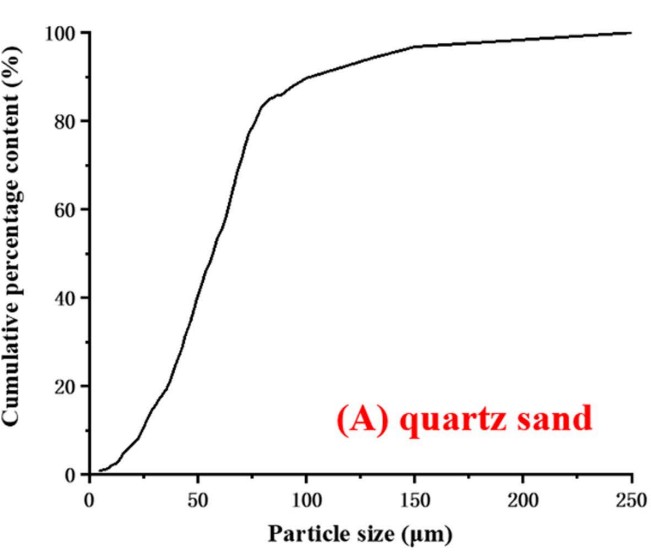
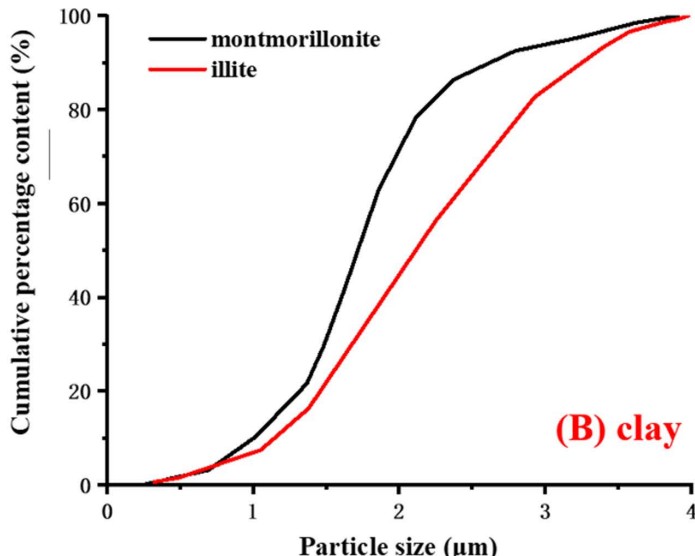

**Fig 3. Relationship between cumulative content and particle size.**

quartz sand and clay with 2ml sodium silicate solution according to the set proportion, fill the mixture into the mold in batches, and gently tamp to achieve preliminary compaction. The sample with a size of φ 25 mm x 50 mm is prepared.

(2) On the triaxial testing machine, conduct core compaction, apply an axial pressure of 10 KN at a speed of 100 N/s. After the axial pressure reaches 10 KN, hold for 20 minutes, then remove the axial pressure and take out the sample.

(3) Use a porosity measurement device to measure the sample porosity.

(4) To determine the hydrate saturation in the sample, add a corresponding quality of saline solution to the sample using the "immersion+drip" composite method. Then, freeze the sample at −2 °C for at least 2 hours.

(5) Two hours before the experiment, the environment is cooled to −2 °C. The sample, deformation sensor and pore pressure holder are assembled in a cold storage, and then loaded into the autoclave to prepare for the NGH generation.

(6) Supply $CH_4$ at the bottom with the pressure of 0.5 MPa to remove residual air from the sample pores. After the exhaust is completed, simultaneously supply $CH_4$ at the top and bottom ends to maintain pressure. Then raise the ambient temperature to 2 °C, increase the confining pressure to 10 MPa, and increase the pore pressure to 9 MPa. Maintain this state for 10 hours. After continuously observing the stability of the airflow meter reading, continue to maintain the system state for 24 hours.

(7) Change the experimental conditions and conduct the in-situ triaxial mechanical experiment. The loading method is displacement controlled, and the loading rate is 0.3 mm/min. Record the failure stress and strain data. The experiment is done.

## 2.4. Experimental scheme

The specific settings of experimental contents are shown in Table 1.

**Table 1. Experimental conditions and basis data of samples.**

| Number | Total clay content | $m_{Montmorillonite}/m_{Illite}$ | Hydrate saturation | Effective confining pressure | Porosity |
|---|---|---|---|---|---|
| 0 | 0% | —— | 0%, 15%, 30%, 45% | 2.5MPa | 34%–36% |
| 1 | 8% | 1:0, 2:1, 1:1, 1:2, 0:1 | 30% | 2.5MPa | 34%–36% |
| 2 | 16% | 1:0, 2:1, 1:1, 1:2, 0:1 | 30% | 2.5MPa | 34%–36% |
| 3 | 24% | 1:0, 2:1, 1:1, 1:2, 0:1 | 30% | 2.5MPa | 34%–36% |
| 4 | 32% | 1:0, 2:1, 1:1, 1:2, 0:1 | 30% | 2.5MPa | 34%–36% |
| 5 | 8% | 1:1 | 0%, 15%, 30%, 45% | 2.5MPa | 34%–36% |
| 6 | 16% | 1:1 | 0%, 15%, 30%, 45% | 2.5MPa | 34%–36% |
| 7 | 24% | 1:1 | 0%, 15%, 30%, 45% | 2.5MPa | 34%–36% |
| 8 | 32% | 1:1 | 0%, 15%, 30%, 45% | 2.5MPa | 34%–36% |
| 9 | 8% | 1:1 | 30% | 0.5MPa, 1MPa, 3MPa, 5MPa | 34%–36% |
| 10 | 16% | 1:1 | 30% | 0.5MPa, 1MPa, 3MPa, 5MPa | 34%–36% |
| 11 | 24% | 1:1 | 30% | 0.5MPa, 1MPa, 3MPa, 5MPa | 34%–36% |
| 12 | 32% | 1:1 | 30% | 0.5MPa, 1MPa, 3MPa, 5MPa | 34%–36% |

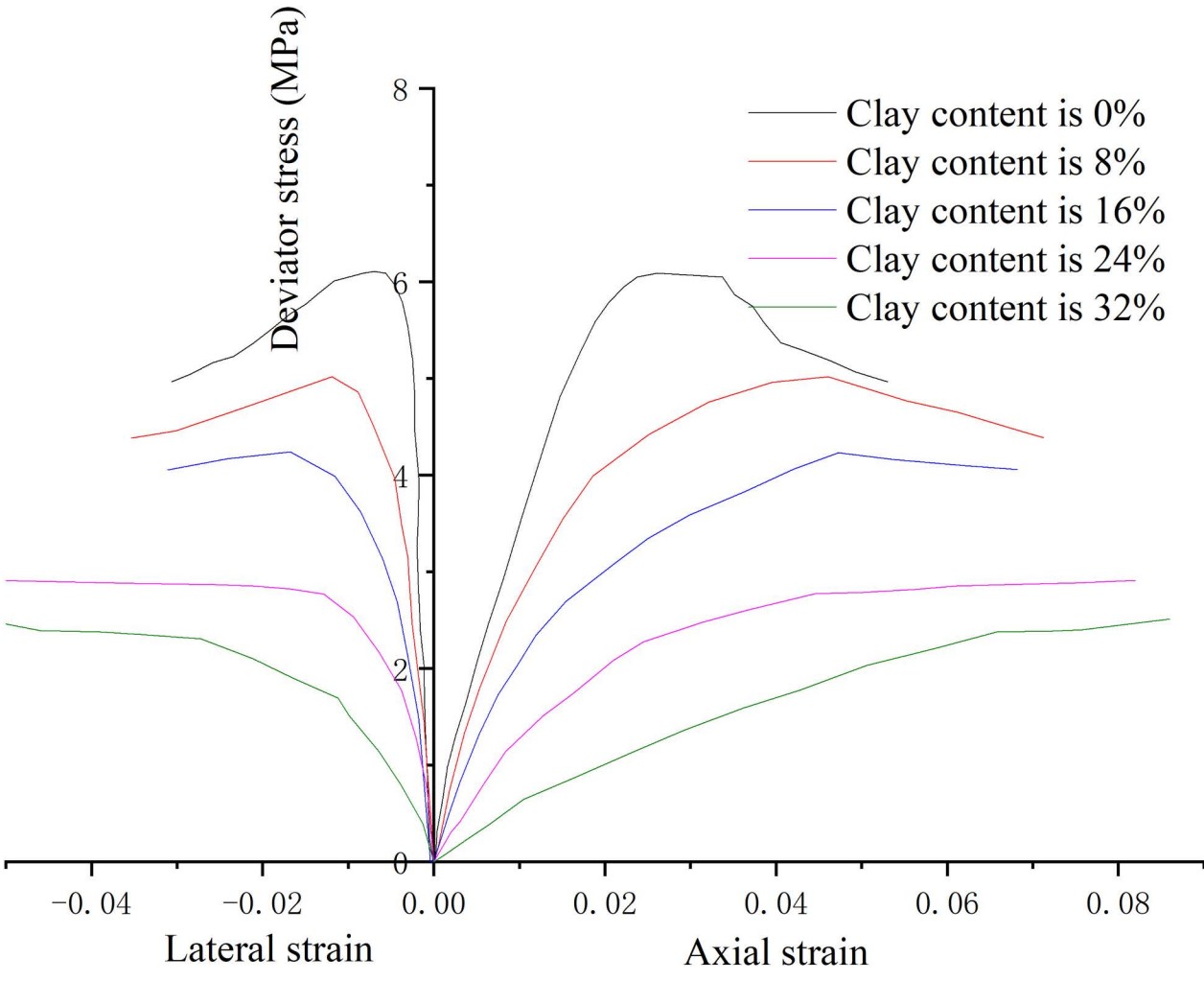

**Fig 4. Stress-strain curves under different clay content.**

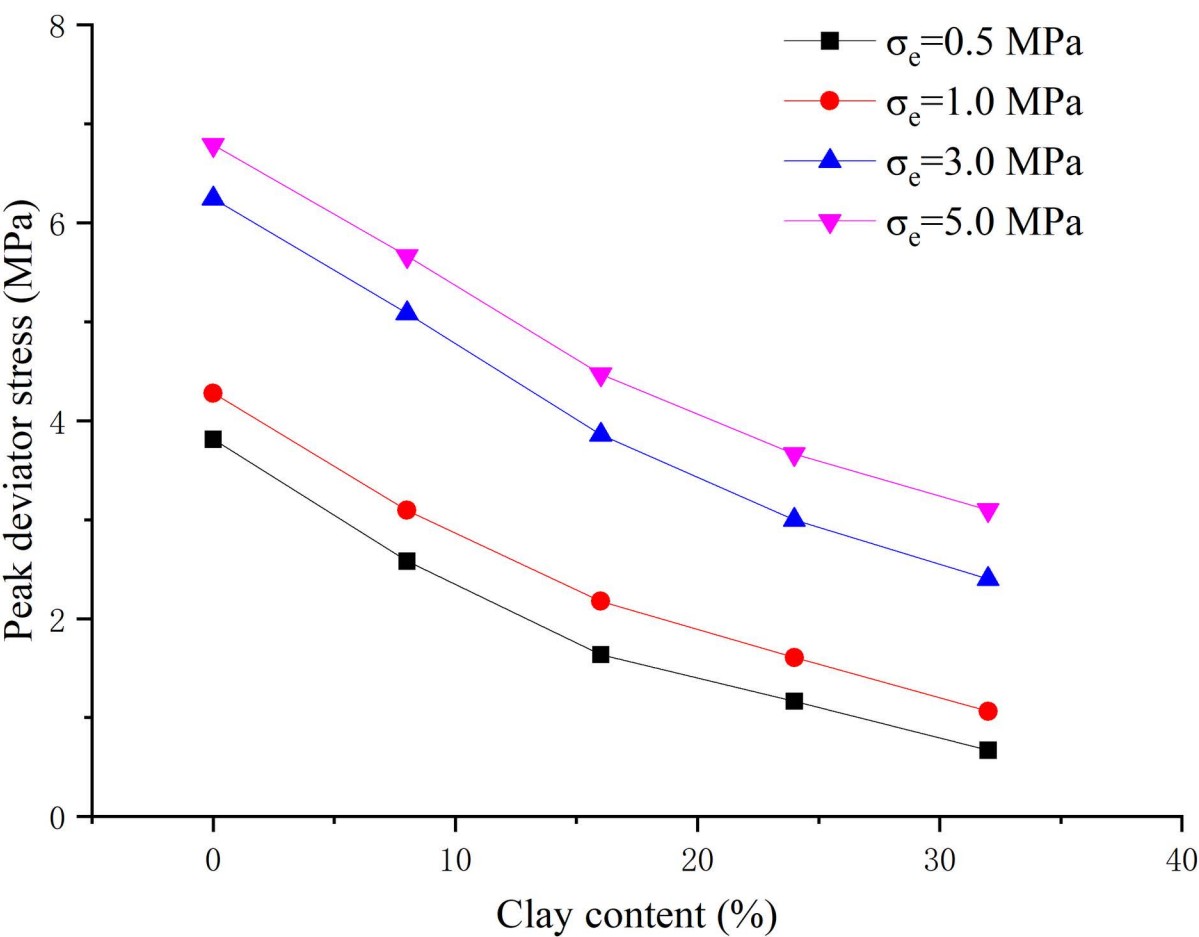

**Fig 5. Relationship between peak strength and clay content (S$_h$ = 30%).**

## 3. Analysis of experimental results

### 3.1. Characteristic analysis of stress-strain curve

According to the hydrate reservoir conditions in Shenhu sea area [49], the test parameters are set. The range of plastic limit values for hydrate-bearing sediment samples with different viscosities is 21.4% –25.6%, the range of liquid limit values is 56.1% –60.3%, and the range of residual moisture content is about 12%. The stress strain curve is obtained when the effective confining pressure is 2.5 MPa, the saturation is 30%, the clay content is 0%, 8%, 16%, 24%, 32% respectively, and the content ratio of montmorillonite/illite is 1:1, as shown in Fig 4.

From Fig 4, the stress-strain curve (C$_{clay}$ = 0%) is obviously divided into compaction stage, elastic stage, yield stage and post peak stage. While the stress-strain of clay hydrate sample is relatively vague in stage division, with short elastic stage and long yield stage. The axial deformation is obviously greater than the radial deformation (axial strain>lateral strain). With the increase of clay content, the peak strength is less obvious, and the strain hardening feature is more prominent. This is because the clay blocks the pore communication channel, hinders hydrate formation, and changes the sediment cementation. Moreover, the clay particles are small in size, easy to be hydrated and expanded, and the friction and cohesion between particles are relatively small, reducing the peak strength. Meanwhile, the slope of the stress-strain

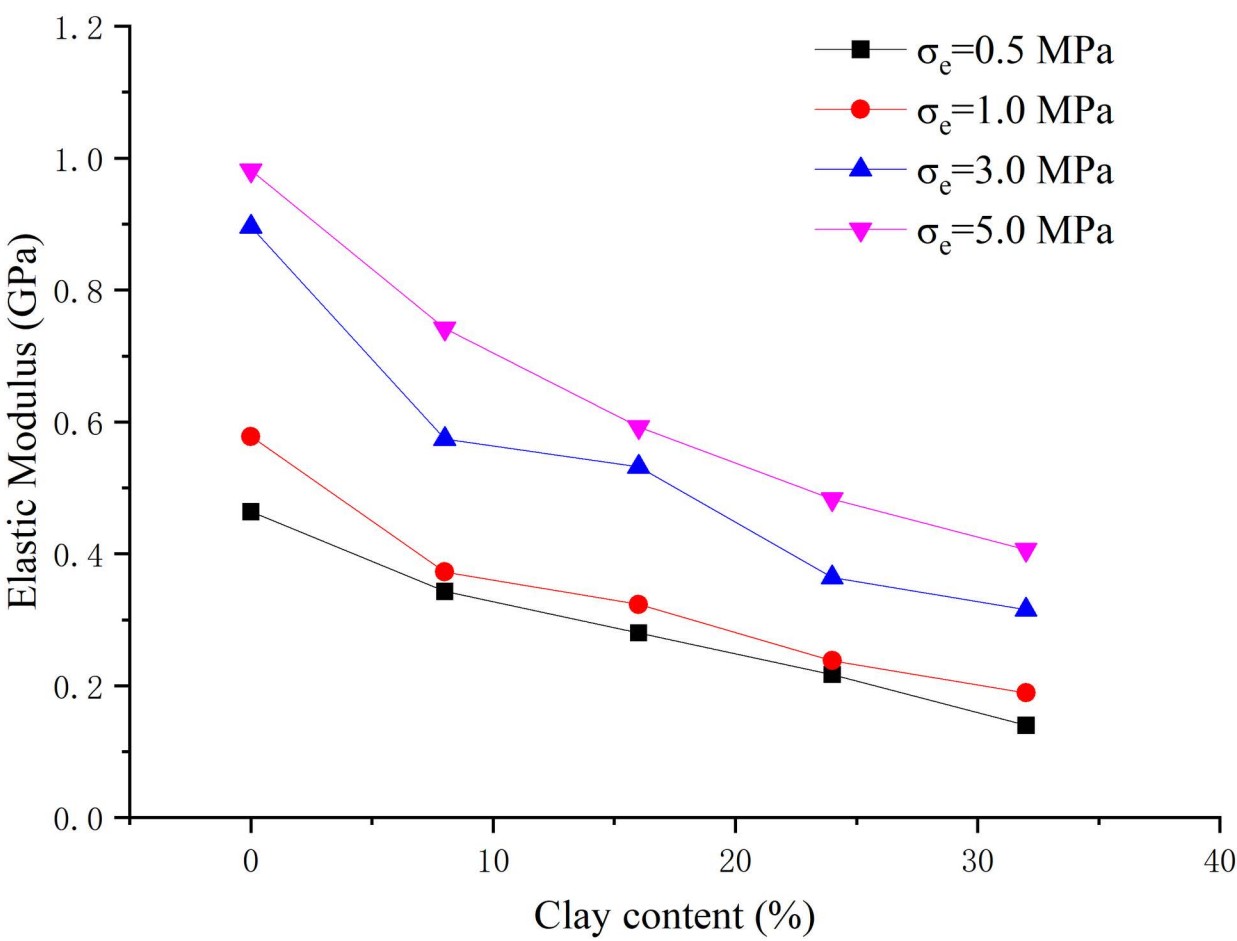

**Fig 6. Relationship between elastic modulus and clay content ($S_h$ = 30%).**

curve decreases, the elastic stage shortens, the plastic stage lengthens, and the deviator stress changes slightly during the plastic stage, indicating that during the loading process, clay particles will enter the hydrate gap, making the sample more compact and bearing capacity enhanced.

### 3.2. Influence law analysis of clay content

When the hydrate saturation is 30%, the effective confining pressure is 0.5 MPa, 1 MPa, 3 MPa, 5 MPa respectively, the clay content is 0%, 8%, 16%, 24%, 32% respectively, and the content ratio of montmorillonite/illite is 1:1, the peak strength, elastic modulus and Poisson's ratio are obtained, as shown in Figs 5–7 respectively.

From Fig 5, with the increase of clay content, the peak strength decreases, but the decrease amplitude is large at the initial stage, then decreases and remains basically unchanged. It is due to the obvious blocking effect on hydrate formation caused by clay entering the pore channel at the initial stage. With the increase of clay content, the peak strength further decreases under the combined effect of clay cementation and pore filling, but the reduction effect is weakened. When the effective confining pressure is large, the effect of clay content on reducing the peak strength will be further weakened. With the

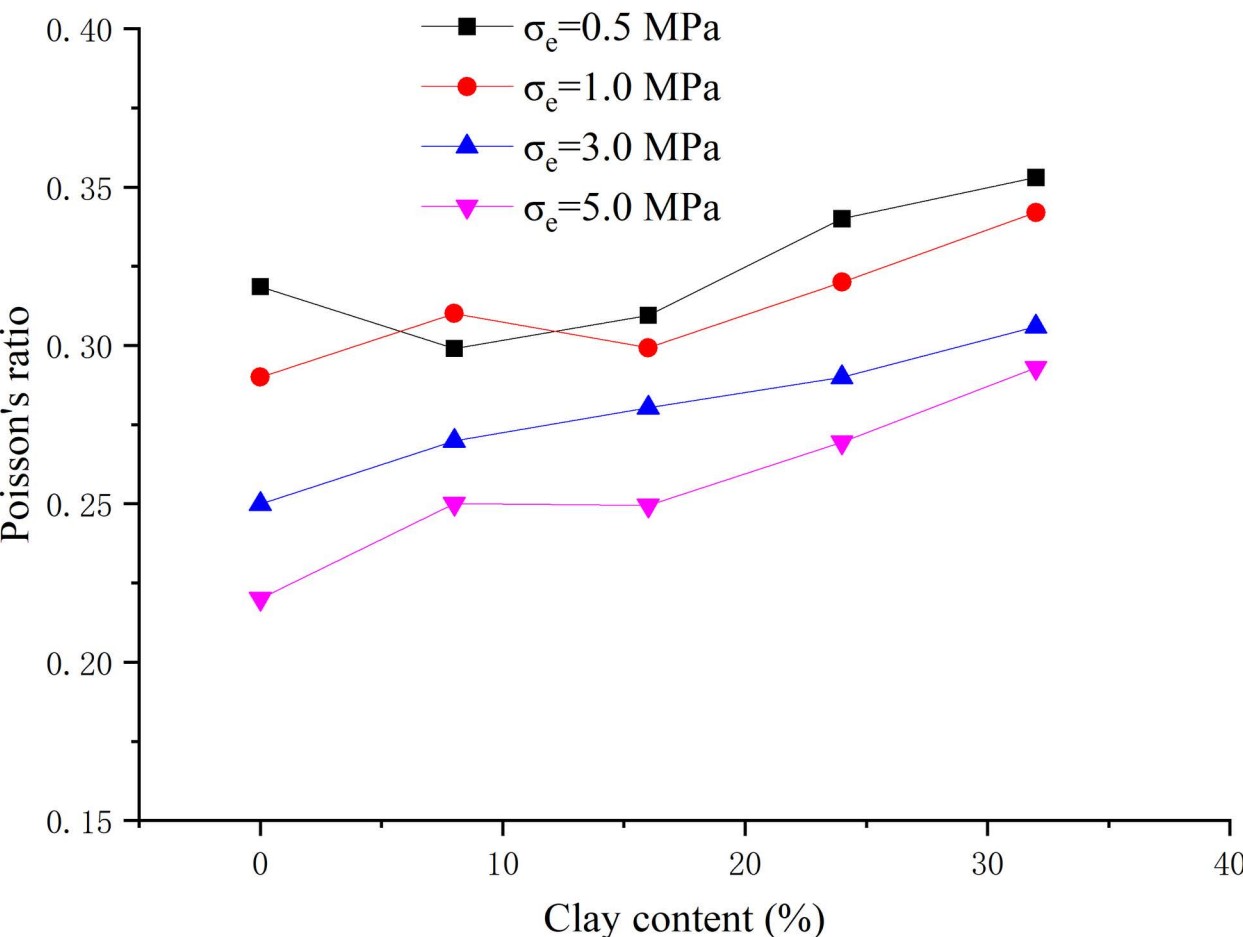

**Fig 7. Relationship between Poisson's ratio and clay content ($S_h$ = 30%).**

increase of effective confining pressure, the peak strength under different clay content shows a nonlinear increase. Especially, when the confining pressure increases from 1 MPa to 3 MPa, the peak strength appears a significant jump. The reason is that the effective confining pressure makes the sample more compact and the strength increases. But when the effective confining pressure increases to a certain extent, the ice crystal would melt under pressure, the clay hydration would be enhanced, and the cementation would be weaken, leading to the strength reduction.

From Fig 6, the elastic modulus is less than 1.0 GPa, showing a strong plasticity. With the increase of clay content, the elastic modulus shows a downward trend, but individual data is irregular. With the increase of effective confining pressure, the elastic modulus increases, and when the confining pressure increases from 1 MPa to 3 MPa, the elastic modulus increases greatly. When the effective confining pressure is low, the elastic modulus changes slightly with clay content. When the effective confining pressure is high, the elastic modulus decreases more obviously with the increase of clay content. From Fig 7, with the variation of clay content, the overall range of Poisson's ratio is 0.2–0.4, showing no regular change. Poisson's ratio decreases with the increase of effective confining pressure generally.

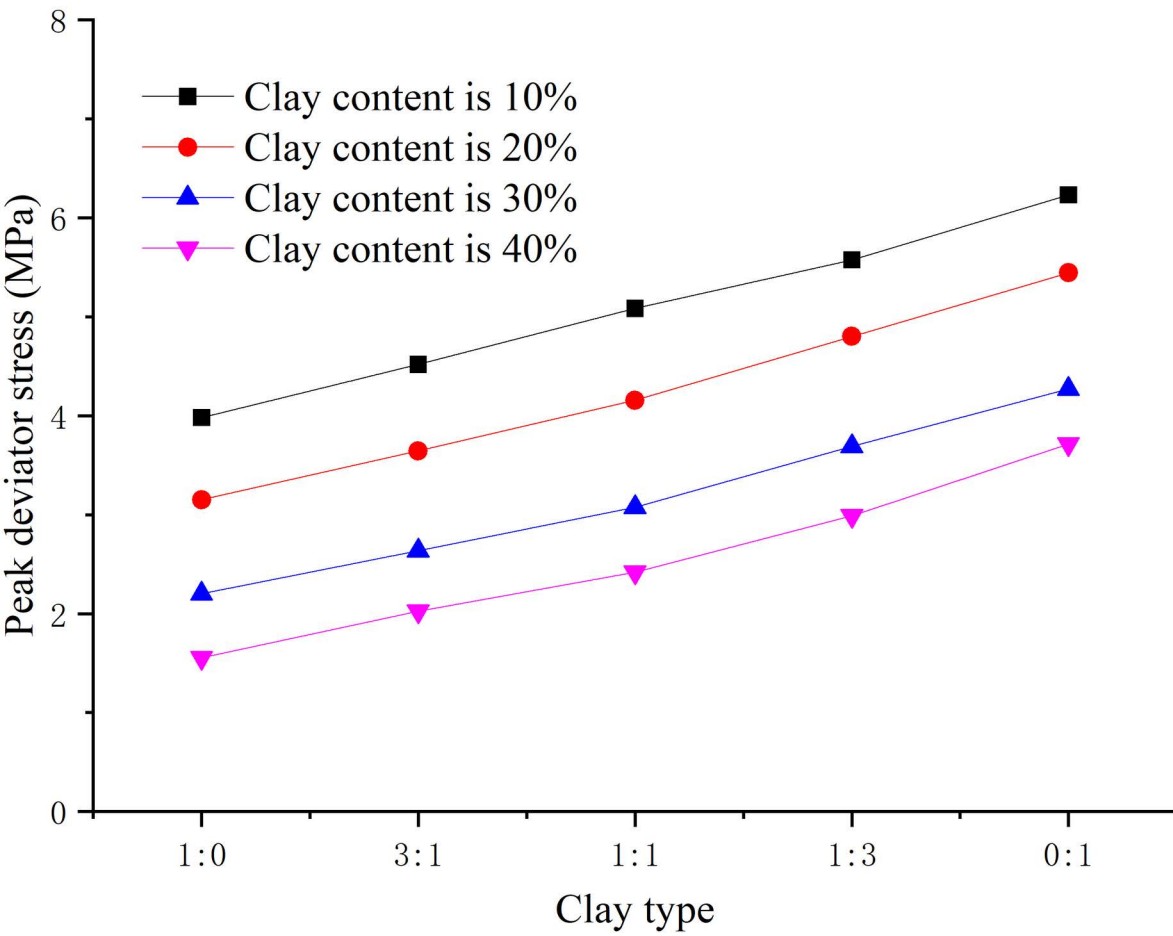

**Fig 8. Relationship between peak strength and clay type ($S_h$ = 30%).**

### 3.3. Influence law analysis of clay type

When the hydrate saturation is 30%, the effective confining pressure is 2.5 MPa, the total clay content is 0%, 8%, 16%, 24%, 32% respectively, and the content ratio of montmorillonite/illite is 1:0, 2:1, 1:1, 1:2, 0:1 respectively, the peak strength, elastic modulus, and Poisson's ratio are obtained, as shown in Figs 8–10 respectively.

From Fig 8, when the content ratio of montmorillonite/illite decreases, the illite content increases, and the peak strength shows an increasing trend, indicating that the cementation of illite is stronger than that of montmorillonite. This is because the particle size of illite is bigger than that of montmorillonite, the specific surface area is smaller, leading to weaker water absorption and swelling properties, the bonding strength of illite crystal layer is greater. Moreover, the hydration film of illite mineral is thinner, and the friction resistance between particles is greater during sliding. However, with the increase of clay content, the peak strength decreases, the illite content increases, and the decrease of peak strength slows down, indicating that clay type influences the peak strength greatly.

From Fig 9, the elastic modulus of hydrate-bearing sediment decreases with the increase of clay content. But with the decrease of montmorillonite/illite content ratio, the illite content increases, and the elastic modulus changes differently, but the overall trend is increasing, and the elasticity is enhanced.

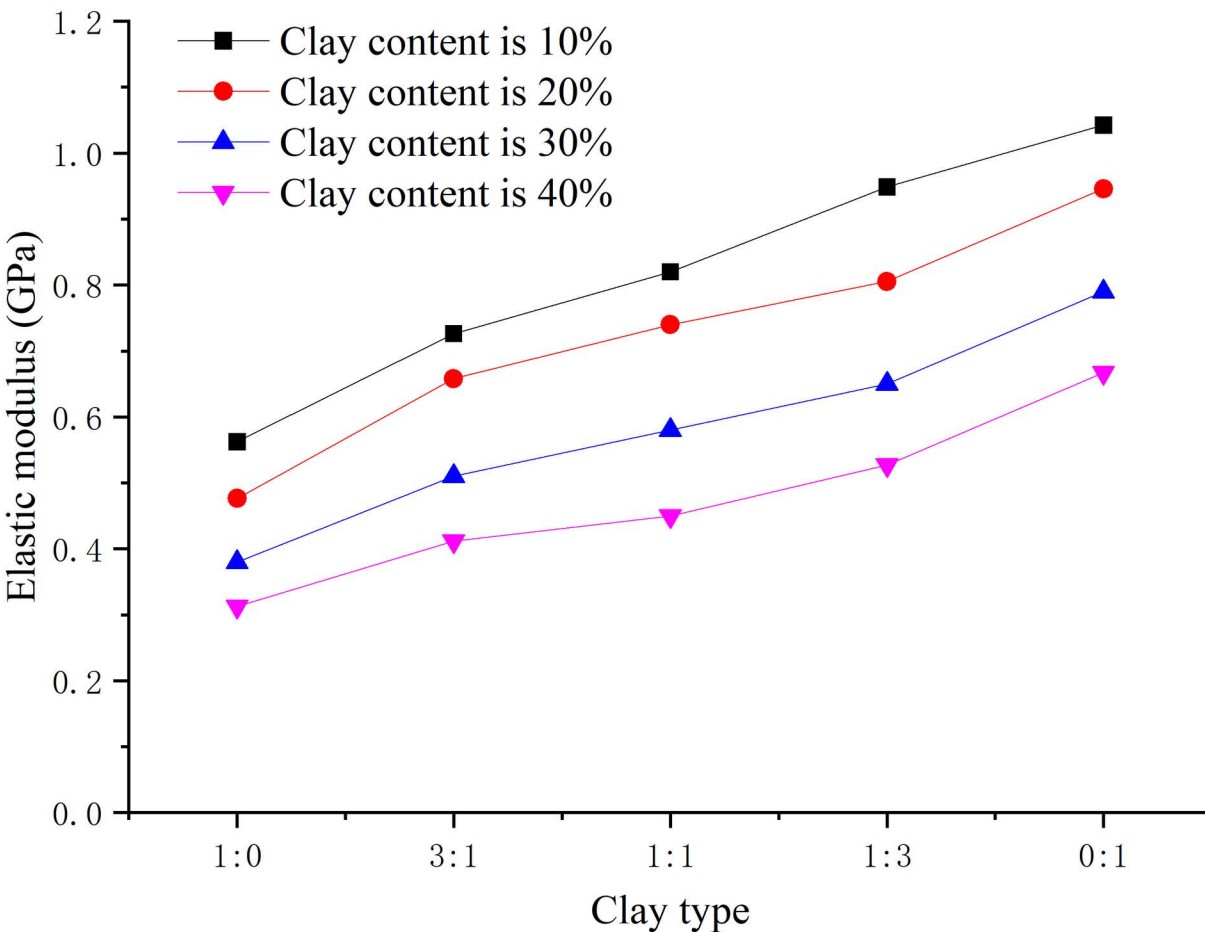

**Fig 9. Relationship between elastic modulus and clay type ($S_h = 30\%$).**

From Fig 10, the Poisson's ratio is mainly distributed between 0.15–0.40. Generally, with the increase of clay content, the Poisson's ratio shows a downward trend, indicating that the more the clay content is, the more severe the deformation is. As the content ratio of montmorillonite/illite decreases, the content of illite increases, and the Poisson's ratio shows a downward trend, which may be caused by the greater connecting force of illite crystal layers and the greater energy required for deformation.

### 3.4. Influence law analysis of hydrate depressurization decomposition

The effective confining pressure is 2.5 MPa, and the content ratio of montmorillonite/illite is 1:1, the clay content is 0, 8%, 16%, 24%, 32% respectively, the peak strength, elastic modulus, and Poisson's ratio are obtained when the hydrate saturation decreases from 45% to 0, 15%, 30%, and 45% respectively, as shown in Figs 11–13 respectively.

As shown in Fig 11, when the hydrate saturation is degraded from 45% to low saturation, the peak strength is lower than that of sample with the same saturation simply prepared, which is caused by the destruction of sediment structure during the hydrate decomposition. The hydrate-bearing sediment with different clay content shows similar mechanical properties. With the decrease of hydrate saturation, the peak strength decreases nearly linearly. This is because NGH has the role of cementation and skeleton support, and the hydrate forms

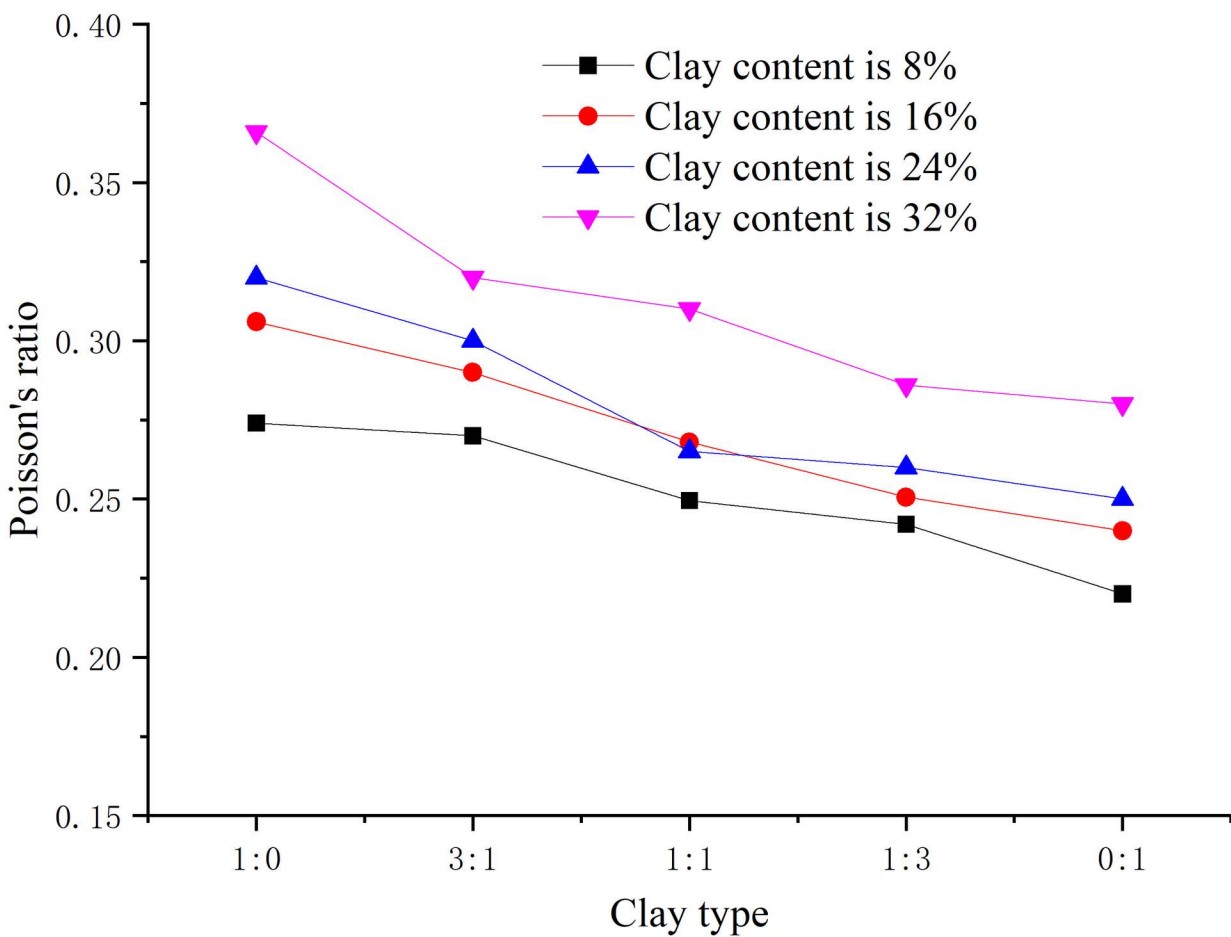

**Fig 10. Relationship between Poisson's ratio and clay type ($S_h = 30\%$).**

adhesion on the particle surface, making the loose contact more solid. When the clay content is high, with the increase of hydrate saturation, the peak strength increases, but the increase amplitude decreases, indicating that the cementation effect and skeleton support effect of clay minerals are weaker than that of hydrate.

From Fig 12, with the increase of hydrate saturation, the elastic modulus increases, but the regularity becomes worse. When the clay content is between 0%–8%, the elastic modulus changes more obviously with the saturation. When the clay content increases to 16%–40%, the changing amplitude decreases, and the influence of hydrate saturation decreases. This is because that when the clay content exceeds 16%, hydrate cementation and framework support are greatly weakened, resulting in weakened elasticity and strengthened plasticity. With the increase of clay content, the elastic modulus decreases, especially when the clay content increases from 0% to 8%. From Fig 13, there is no clear law between Poisson's ratio and hydrate saturation, but in general, with the increase of hydrate saturation and decrease of clay content, Poisson's ratio decreases, fluctuating between 0.2–0.4.

### 3.5. Influence law analysis of effective confining pressure

The hydrate saturation is 30%, the clay content is 0%, 8%, 16%, 24%, 32% respectively, and the effective confining pressure is 0.5 MPa, 1.0 MPa, 3.0 MPa and 5.0 MPa respectively. The

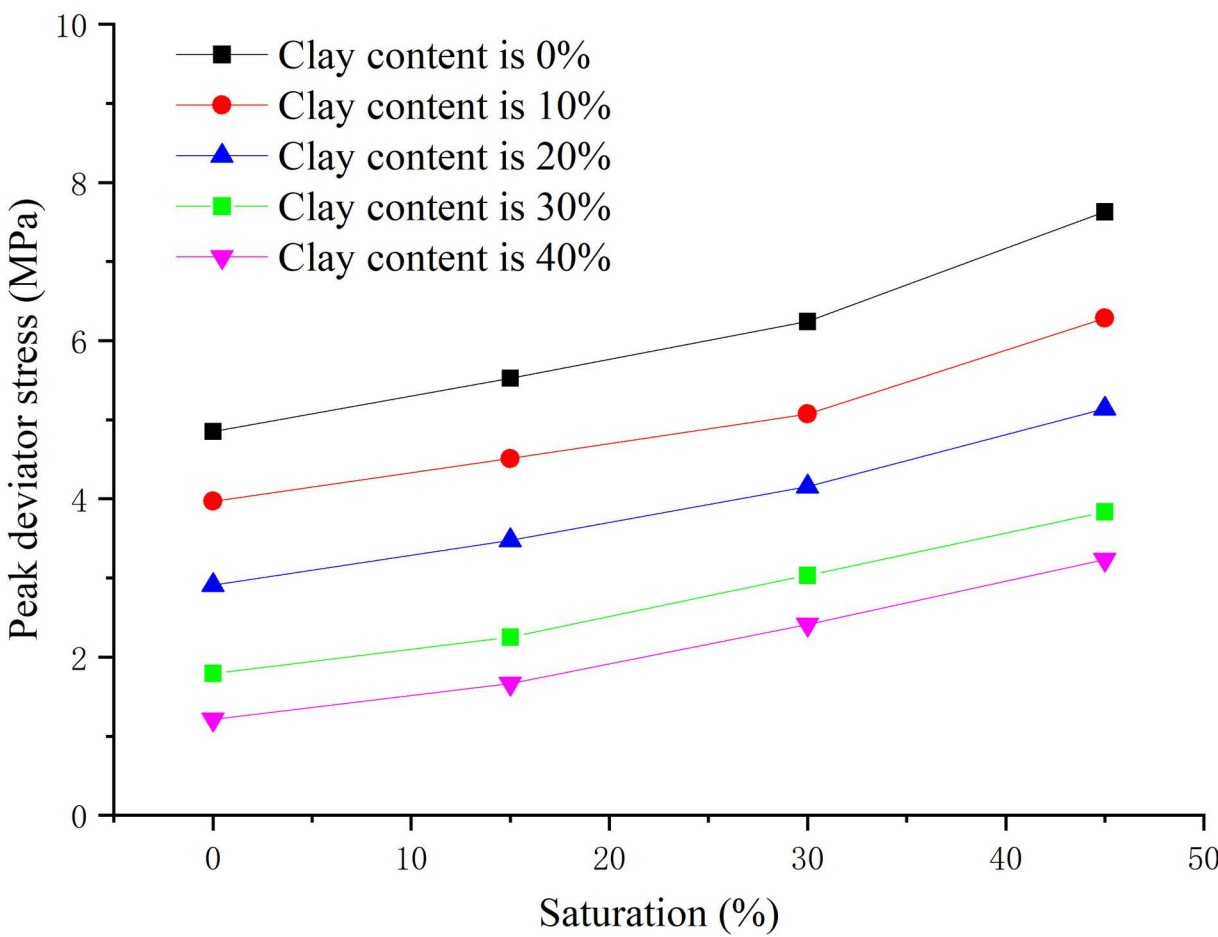

**Fig 11. Relationship between peak strength and hydrate saturation.**

peak strength, elastic modulus and Poisson's ratio are obtained as shown in Figs 14–16 respectively.

From Fig 14, when the clay content is certain, the peak strength increases with the increase of effective confining pressure. The peak strength increased greatly when the effective confining pressure is within 0–3 MPa, while small when the effective confining pressure is 3–5 MPa. When the effective confining pressure remains unchanged, the peak strength decreases with the increase of clay content, but the decreasing amplitude decreases. This is because the increase of effective confining pressure enhances the binding effect on sediment particles, increases the biting force and density between sediment particles, and increases the frictional resistance for relative movement.

From Fig 15, the elastic modulus increases with the increase of effective confining pressure. When the effective confining pressure is less than 3 MPa, the elastic modulus increases greatly, while small when the effective confining pressure is greater than 3 MPa. When the effective confining pressure remains unchanged, the elastic modulus decreases with the increase of clay content, but the reduction extent decreases. From Fig 16, Poisson's ratio has no obvious correlation with effective confining pressure. In general, Poisson's ratio decreases with the increase of effective confining pressure, indicating the sample is laterally restrained caused by effective confining pressure at the stage of axial loading.

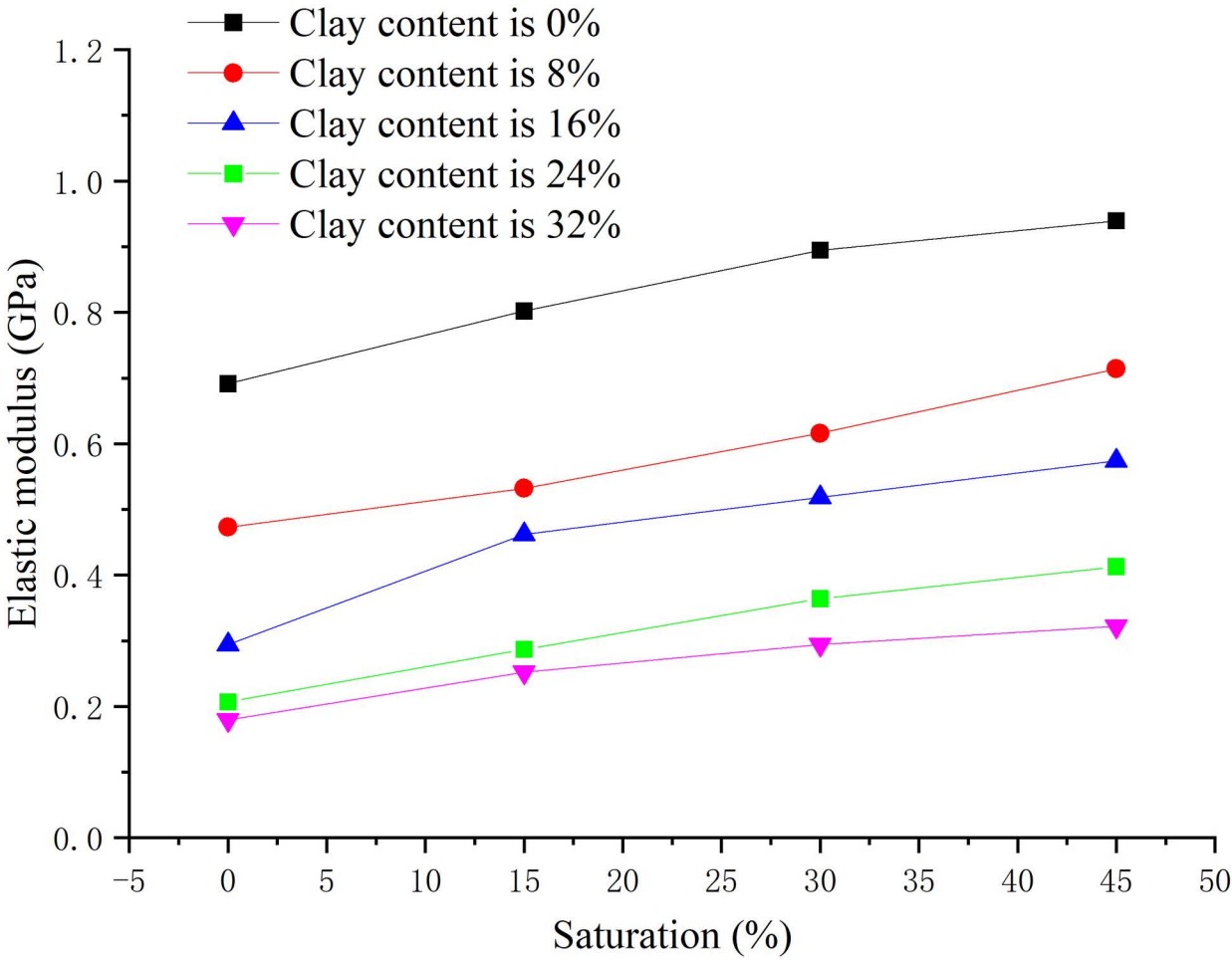

**Fig 12. Relationship between elastic modulus and saturation of hydrate-bearing sediment.**

## 4. Discussions

Clay mineral is an important component of shallow hydrate-bearing sediment in the Shenhu sea area, with a content of 0–36%, which influences the mechanical properties greatly. Currently, there are many studies on the mechanical properties and influence factors, but few studies on clay content and clay type, which would have impact on well-bore collapse, reservoir sand production and fracturing stimulation during drilling and exploration.

Therefore, through in-situ synthesis of artificial samples similar to the mineral components of shallow unconsolidated hydrate in the Shenhu sea area, mechanics experiments are carried out to explore the mechanical properties of clay hydrate-bearing sediments, especially the influence of clay content and clay type. Currently, a large number of studies have been conducted on the mechanical properties of various types of hydrate-bearing sediments (fine sandy, silty), the effects of hydrate saturation, effective confining pressure, temperature are focused on. While there are few experimental studies on the impact of clay content and clay type, and this research can better compensate for this lack, which will have great significance for guiding hydrate drilling and exploration operations.

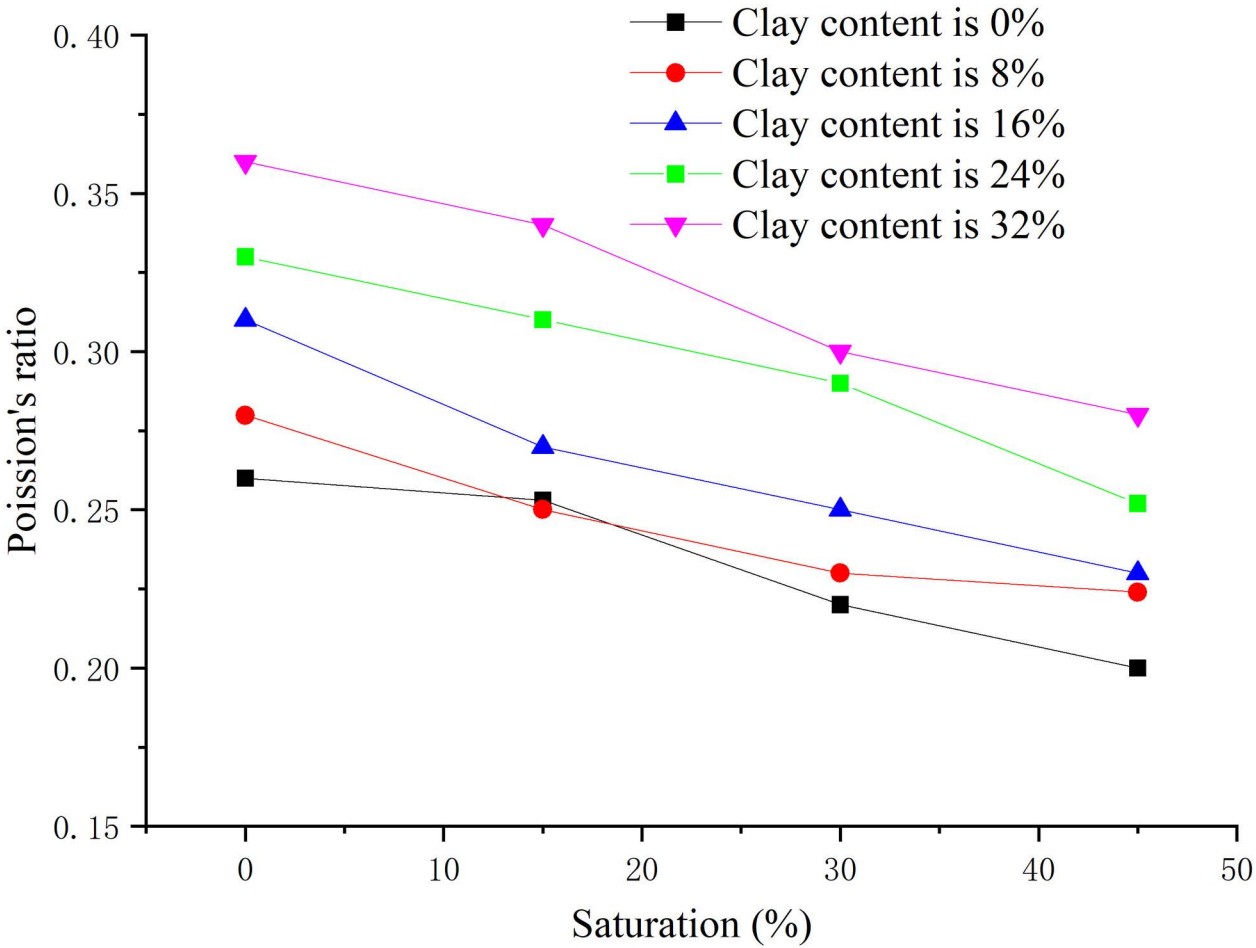

**Fig 13. Relationship between Poisson's ratio and saturation of hydrate-bearing sediment.**

The montmorillonite and illite are discussed in the paper, and the mineral components are shown in Table 2. The purity of montmorillonite is 88.87%, and the purity of illite is 89.08%, both are high. From the comparison between Fig 3A and 3B, the particle size of selected montmorillonite and illite is less than 5 μm, which is significantly smaller than the size of quartz sand. And the particle size of montmorillonite is smaller than that of illite. The clay causes changes in the structure of sediment. As the clay content increases, the average particle size of the sediment gradually decreases, thereby the frictional force between particles is reduced and the sample strength decreases also. At the same time, clay minerals themselves also have a certain lubricating effect on sediment particles, which can further weaken the friction between particles. Therefore, the higher the clay content, the lower the peak strength compared to sandy hydrate. Moreover, because the particle size of montmorillonite is smaller than illite, the specific surface area is larger, the water absorption capacity is stronger, and the water film formed on the particle surface is thicker, resulting in less contact between particles, less friction and cohesion, more prone to dislocation of particles, and rock damage. Therefore, when the montmorillonite content is high, the hydrate-bearing sediment shows reduced strength and enhanced plasticity.

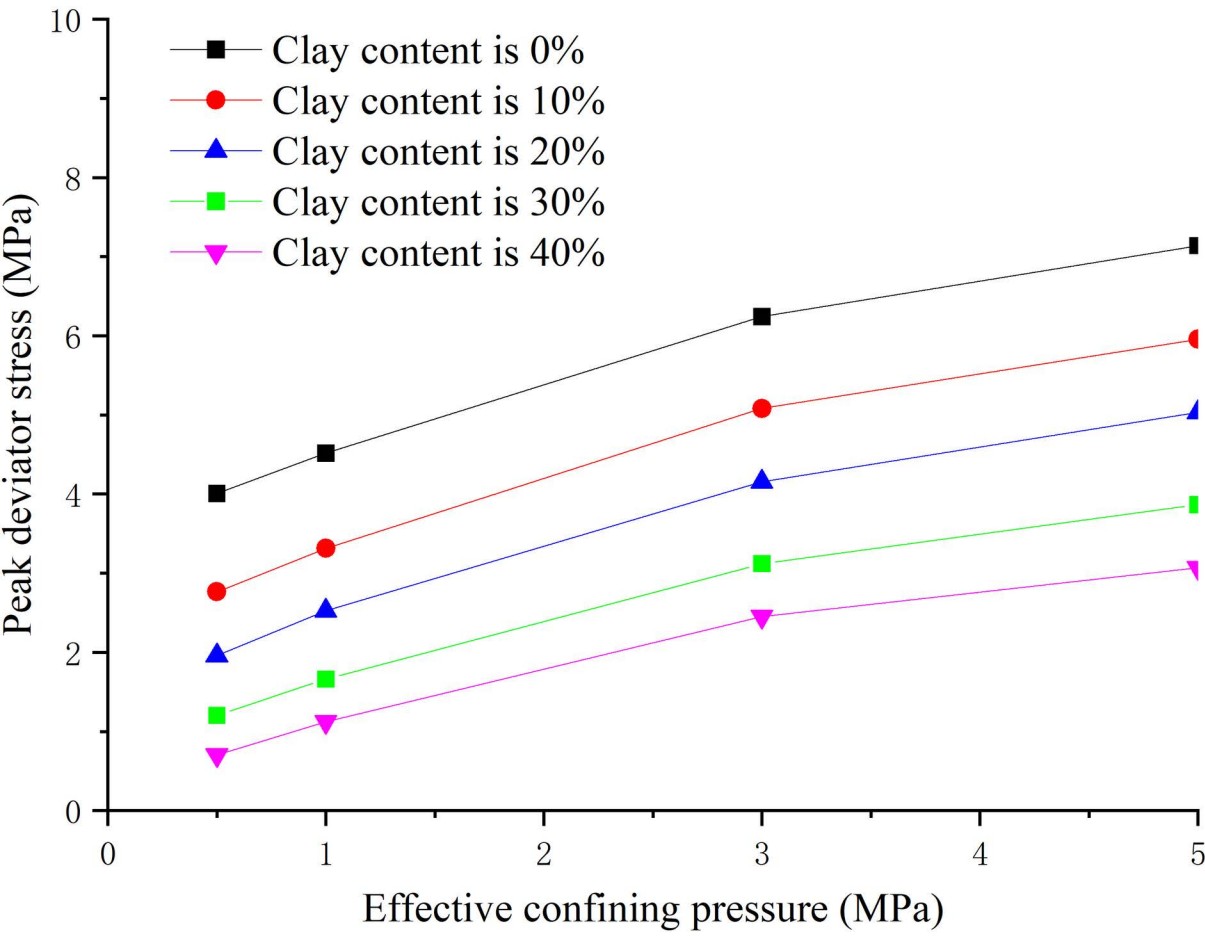

**Fig 14.  Relationship between peak strength and effective confining pressure ($S_h$ =30%).**

Due to limited time and article space, some work needs to be further carried out. We have obtained the change rules and influencing factors of mechanical properties, especially the clay content and clay type, and have provided theoretical explanations for the change rules. However, there is a lack of support from the micro structure level, and a more in-depth exploration of the structure and mechanism of the two clay minerals-montmorillonite and illite needs to be conducted. In the future, SEM technology and CT technology will be developed, which will be helpful to explain the microscopic mechanism of clay mineral action. The qualitative introduction of the mechanical properties changes under the effect of clay minerals is foused on, but in-depth analysis and summary are not provided. The formation of failure strength criteria and constitutive models for hydrated sediments that consider the impact of clay minerals limits the application of research results, which will be the next research direction.

## 5.  Conclusions

Fully considering the composition and geological conditions of shallow clay hydrate in Shenhu sea area, NGH samples with similar composition are synthesized in situ, and triaxial experiments are carried out to analyze the mechanical properties of unconsolidated

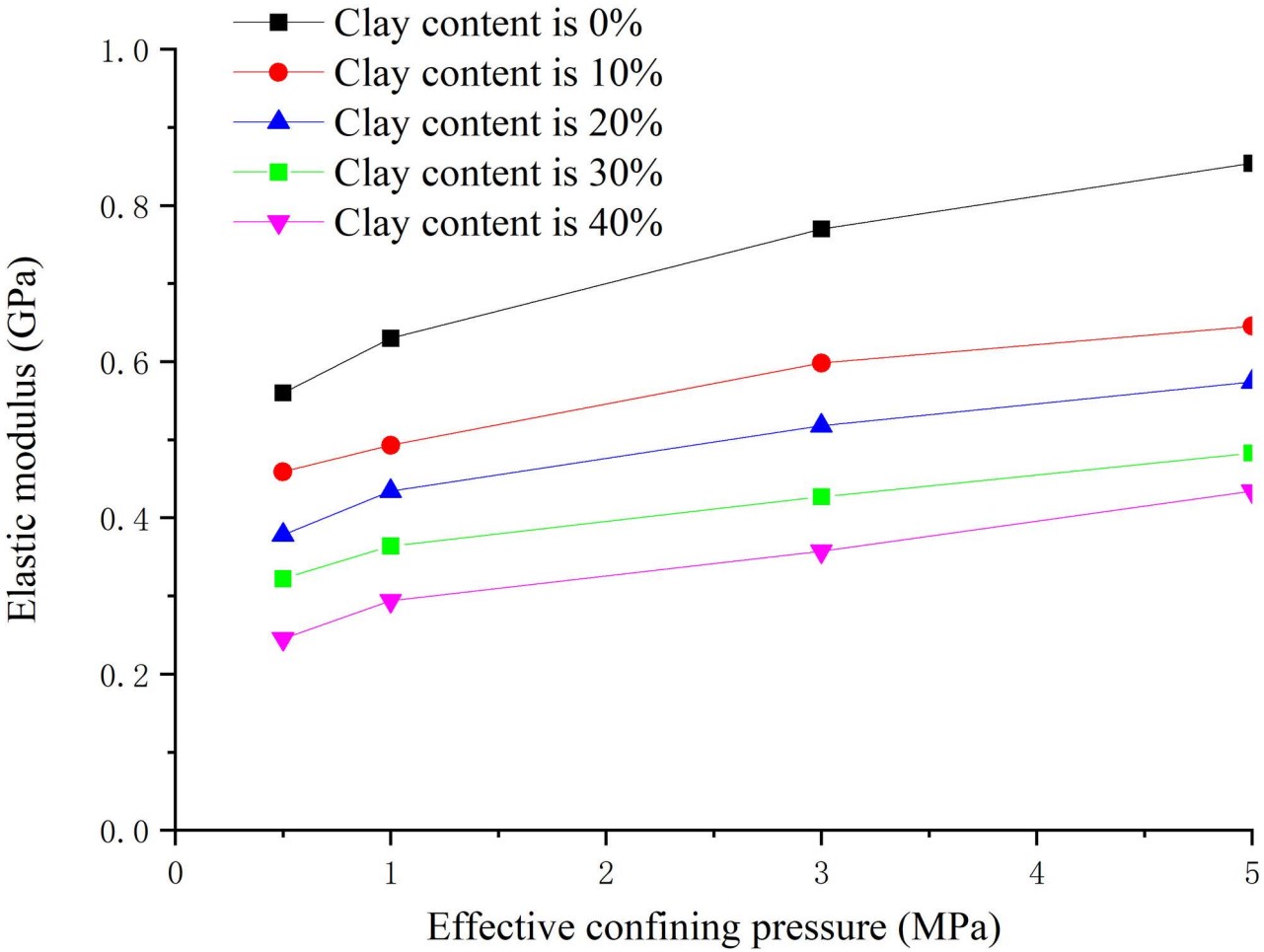

**Fig 15.  Relationship between elastic modulus and effective confining pressure ($S_h = 30\%$).**

hydrate-bearing sediment under different conditions. The following conclusions are obtained:

(1) Different from silty hydrate, the strain hardening characteristics of clay silt hydrate are more prominent. With the increase of clay content, the peak strength is less obvious, the plasticity is enhanced. This is because the clay blocks the pore connectivity channel, hinders hydrate formation and changes the sediment cementation.

(2) The peak strength, elastic modulus and Poisson's ratio show a downward trend with the increase of clay content, but the peak strength and elastic modulus change more obviously. Under the influence of hydrate saturation and effective confining pressure, hydrate-bearing sediment with different clay content shows similar mechanical laws. The peak strength changes linearly with hydrate saturation, while nonlinearly with effective confining pressure, especially 0–3 MPa.

(3) When the content ratio of montmorillonite/illite decreases, the peak strength and elastic modulus show an increasing trend. Because compared with montmorillonite, the frictional resistance and connection strength of illite crystal layer are larger with bigger particle size, weaker hydration and thinner water film.

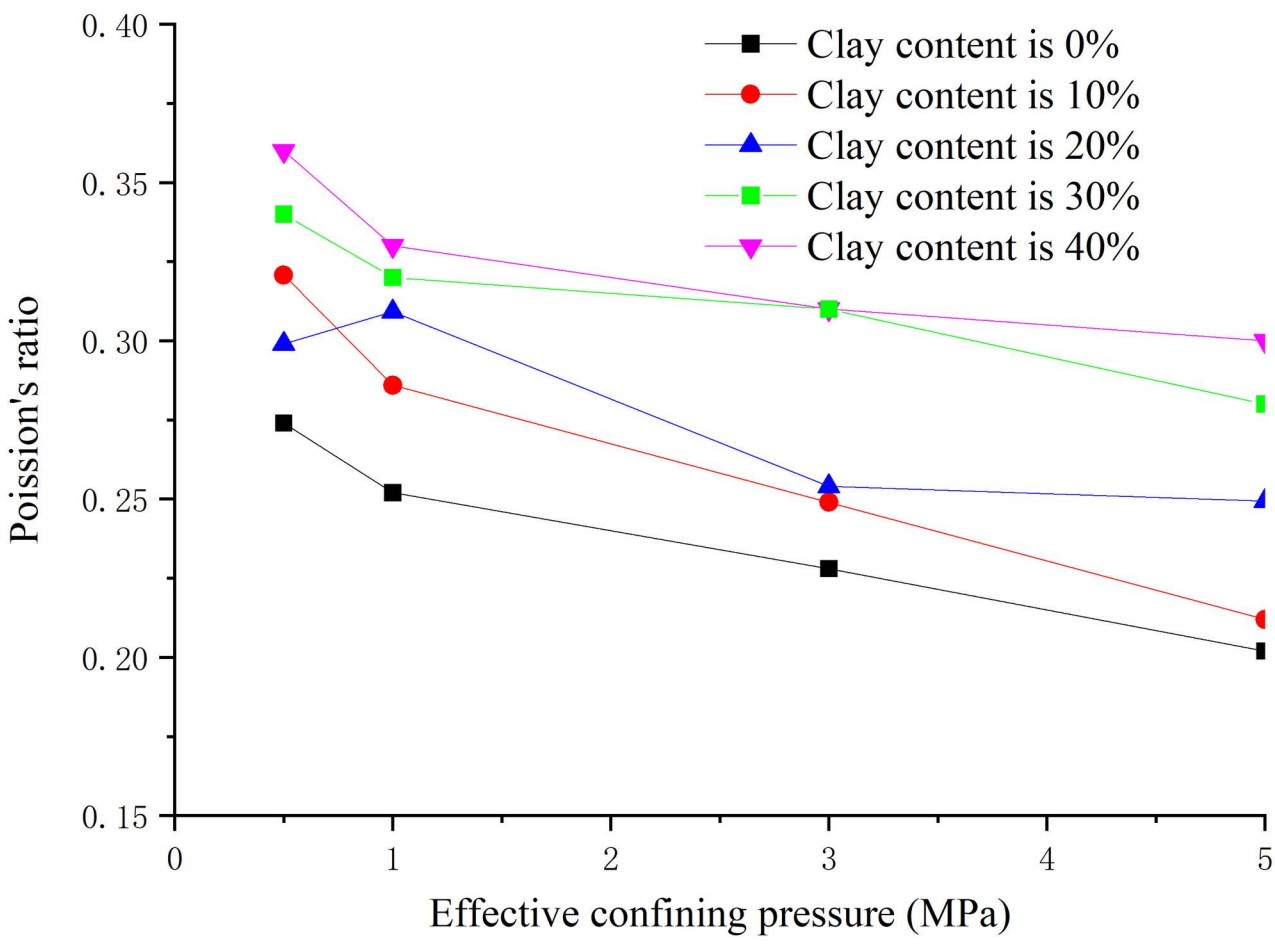

**Fig 16. Relationship between Poisson's ratio and effective confining pressure ($S_h = 30\%$).**

**Table 2. Clay mineral components.**

| Minerals | Montmorillonite | Illite | Quartz | Calcite | Potassium feldspar | Plagioclase |
|---|---|---|---|---|---|---|
| Montmorillonite mineral | 88.87% | —— | 5.91% | 2.86% | 0.52% | 1.84% |
| Illite mineral | —— | 89.08% | 6.09% | 2.65% | 0.64% | 1.54% |

## Author contributions

**Conceptualization:** Yuanfang Cheng.

**Data curation:** Yuanwei Sun.

**Formal analysis:** Yuanfang Cheng, Liqiang Wang.

**Funding acquisition:** Chuanliang Yan.

**Investigation:** Cui Li, Xiaodong Dai.

**Methodology:** Yuanwei Sun.

**Resources:** Yuanwei Sun.

**Writing – original draft:** Yuanwei Sun.

**Writing – review & editing:** Yuanwei Sun.

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
