## [Decision Letter · Decision Letter 0]

9 Sep 2024

PONE-D-24-35292Mechanical Properties Research of Unconsolidated Hydrate-Bearing Sediments under the Effect of Clay MineralsPLOS ONE

Dear Dr. Sun,

Thank you for submitting your manuscript to PLOS ONE. After careful consideration, we feel that it has merit but does not fully meet PLOS ONE’s publication criteria as it currently stands. Therefore, we invite you to submit a revised version of the manuscript that addresses the points raised during the review process.

**ACADEMIC EDITOR: Please insert comments here and delete this placeholder text when finished.** Be sure to:

Please carefully read and address the comments from two reviewers.Improve the quality of the manuscript in both English and presentations.

We look forward to receiving your revised manuscript.

Kind regards,

Jianguo Wang, PhD

Academic Editor

PLOS ONE

Journal Requirements:

 This work had been financially supported by the Dongying Science Development Fund (DJ2023001), the National Natural Science Foundation Project of China (51974353, 51991362, 52104014), the Natural Science Foundation of Shandong Province (ZR2019ZD14), the CNPC Major Science and Technology Project (ZD2019−184−003) and the Project Establishment and Construction Team of Young and Innovative Talents Introduction and Education Plan of Colleges and Universities in Shandong Province. 

This work had been financially supported by the Dongying Science Development

Fund (DJ2023001), the National Natural Science Foundation Project of China

(51974353, 51991362, 52104014), the Natural Science Foundation of Shandong

Province (ZR2019ZD14), the CNPC Major Science and Technology Project

(ZD2019−184−003) and the Project Establishment and Construction Team of Young

and Innovative Talents Introduction and Education Plan of Colleges and Universities

in Shandong Province.

 This work had been financially supported by the Dongying Science Development Fund (DJ2023001), the National Natural Science Foundation Project of China (51974353, 51991362, 52104014), the Natural Science Foundation of Shandong Province (ZR2019ZD14), the CNPC Major Science and Technology Project (ZD2019−184−003) and the Project Establishment and Construction Team of Young and Innovative Talents Introduction and Education Plan of Colleges and Universities in Shandong Province.

6. Please amend the manuscript submission data (via Edit Submission) to include author Liqiang Wang, Xiaodong Dai, Chuanliang Yan.

Reviewers' comments:

Reviewer's Responses to Questions

**Comments to the Author**

1. Is the manuscript technically sound, and do the data support the conclusions?

Reviewer #1: Partly

Reviewer #2: Yes

2. Has the statistical analysis been performed appropriately and rigorously? 

Reviewer #1: No

Reviewer #2: Yes

3. Have the authors made all data underlying the findings in their manuscript fully available?

Reviewer #1: Yes

Reviewer #2: Yes

4. Is the manuscript presented in an intelligible fashion and written in standard English?

Reviewer #1: Yes

Reviewer #2: Yes

5. Review Comments to the Author

Reviewer #1: The authors tried to explore the effect of clay minerals on mechanical properties of hydrate-bearing silt-clay mixture according to many triaxial shearing tests. However, determination of the hydrate saturation has not been clearly introduced, and I guess the authors calculate the hydrate saturation by assuming all the pore water could be formed as methane hydrate. If so, there will be obvious errors especially when the specimen is consisted of fine sands and clays. As we all know that clay minerals have a strong capability to absorb water at its surface, and these water molecules are constrained by electric force. Under this condition, quite a lot of pore water could not react with methane gas to form methane hydrate. This could also explain what has been shown in the figures. For example, Figure 12, when the clay content is high, indicating intensive clay absorbed water in pores, methane hydrate saturation is low when subjected to the same mixed water in soils. This could also weaken he elastic modulus. The third conclusion “when the content ratio of montmorillonite/illite decrease, the peak strength and elastic modulus increase”. This could be also a result of that montmorillonite has a stronger capability to absorb water molecules than illite. The authors should carefully check with this point.

The authors summarized the basic knowledge related to the manuscript topic and found a gap that “researches … fail to consider NGH decomposition, clay content and clay type.” However, related effort has been reported, and here just name a few published papers that the authors may consult. Jiang et al., (2024), Advances in Geo-Energy Research, 11(1): 41-53; Li et al., (2021), Advances in Geo-Energy Research, 5(1): 75-86; Zhang et al., (2024), Measurement, 238, 115369.

Equation (1): The left-hand side is a pressure dimension. However, the dimension of the right-hand side is unknown.

Figure 3: For clays, plastic and liquid limits are much more important than the grain size distribution. Thus, testing data of the limits should be added, and the overall grain size distribution of clay-silt mixtures will be better than the separated grain size distribution.

Reviewer #2: The present paper intends to explore the mechanical properties of unconsolidated hydrate sediments, analyze variation laws and underlying reasons by considering hydrate saturation, effective confining pressure, clay content and clay type. This is helpful for wellbore instability analysis and sand production prediction, which is very interesting and meaningful. However, there are still few questions need to be clarified and discussed.

1. Common clay types include kaolinite, montmorillonite, illite and chlorite. This paper mainly discusses montmorillonite and illite, which are believed the main composition of shallow clay minerals in Shenhu sea area. What about kaolinite and chlorite? The type, content and distribution characteristics of clay minerals in Shenhu sea area should be supplemented detailedly in the introduction.

2. Some details of the experiment need to be discussed. Are the gas hydrate samples prepared and tested in one cell or in different? What is the initial water saturation? What methods are used to prepare uniform samples?

3. In the paper, there is no information on the place where the temperature is measured. In addition, temperature has a significant impact on the mechanical properties of hydrate sediment. The higher the clay content, the more obvious the effect may be. What is the temperature variation during the mechanical testing? Would this variation have an impact on test results and model establishment? The effect of temperature on unconsolidated hydrate sediments is not considered in this paper, and can be the future research direction (I share a good idea and conception with the Authors).

4. In the experiment, how to get the desired hydrate saturation by hydrate depressurization decomposition? Does the hydrate samples remain uniform during the process?

5. This paper gives the influence of two clay minerals, montmorillonite and illite, on the mechanical properties and strength parameters, which clay mineral has greater influence? Why?

6. The latest references are listed in the paper. However, there are few format issues in the reference. Carefully proofread the reference and unify the author's name citation format.

6. PLOS authors have the option to publish the peer review history of their article (what does this mean? ). If published, this will include your full peer review and any attached files.

**Do you want your identity to be public for this peer review?** For information about this choice, including consent withdrawal, please see our Privacy Policy .

Reviewer #1: No

Reviewer #2: No

---

## [Author Response · Author response to Decision Letter 0]

10 Jan 2025

Reviewer #1:

1.The authors tried to explore the effect of clay minerals on mechanical properties of hydrate-bearing silt-clay mixture according to many triaxial shearing tests. However, determination of the hydrate saturation has not been clearly introduced, and I guess the authors calculate the hydrate saturation by assuming all the pore water could be formed as methane hydrate. If so, there will be obvious errors especially when the specimen is consisted of fine sands and clays. As we all know that clay minerals have a strong capability to absorb water at its surface, and these water molecules are constrained by electric force. Under this condition, quite a lot of pore water could not react with methane gas to form methane hydrate. This could also explain what has been shown in the figures. For example, Figure 12, when the clay content is high, indicating intensive clay absorbed water in pores, methane hydrate saturation is low when subjected to the same mixed water in soils. This could also weaken he elastic modulus. The third conclusion “when the content ratio of montmorillonite/illite decrease, the peak strength and elastic modulus increase”. This could be also a result of that montmorillonite has a stronger capability to absorb water molecules than illite. The authors should carefully check with this point.

Answer: 1. Dear reviewer, the viewpoint is very reasonable and our previous considerations are not very enough. Accurately obtaining hydrate saturation is somewhat difficult. Previously, we have been using saturated water or saturated gas methods to prepare hydrates in order to obtain hydrate saturation more accurately. The first method is adopted in this paper. However, as you mentioned, water will be adsorbed into clay, so the amount of hydrates is not accurated enough. In the experiment, the hydrate saturation is controlled by the mass of water in the sample, and the actual hydrate saturation is measured. While the amount of hydrate formation is modified by measuring the CH4 volume after the experiment. If the amount of supplementary gas is used to calculate the amount of hydrate formation, there are also certain difficulties in accurately measuring the amount of gas. Therefore, we comprehensively consider the gas volume to obtain the hydrate saturation as accurately as possible. See lines 133-135 on Page 5.

2.The authors summarized the basic knowledge related to the manuscript topic and found a gap that “researches … fail to consider NGH decomposition, clay content and clay type.” However, related effort has been reported, and here just name a few published papers that the authors may consult. Jiang et al., (2024), Advances in Geo-Energy Research, 11(1): 41-53; Li et al., (2021), Advances in Geo-Energy Research, 5(1): 75-86; Zhang et al., (2024), Measurement, 238, 115369.

Answer: 2.Thank you very much for the suggestions. After carefully reading these papers, I gain a lot. There is some difference. The decomposition of hydrates is considered, however the content of clay and the types of clay are not considered enough in these papers.

3.Equation (1): The left-hand side is a pressure dimension. However, the dimension of the right-hand side is unknown.

Answer: 3. Li Shuxia et al. conducted natural gas hydrate phase equilibrium experiments in the laboratory using a sand filling model with a porosity of 35%, and obtained data curves as shown in Figure 1.

Fig.1 Equilibrium phase curve of NGH in porous media

Furthermore, through data fitting, equation (1) is obtained as follows:

(1)

Where, P is the pressure of NGH system, MPa; T is the temperature of NGH system, K.

The aqueous solution and porous medium materials used in the experiment are similar to those used in this study, so the natural gas hydrate phase equilibrium model established in the porous medium is chosen as the basis for determining the conditions for the formation of natural gas hydrates in this study.

As shown in Figure 1, when the pore pressure is 9MPa, methane hydrate can theoretically be generated at temperatures below 12.36 ℃. But in order to reduce reaction time, the experimental temperature is set at 2 ℃. See lines 117-122 on Page 5.

4. Figure 3: For clays, plastic and liquid limits are much more important than the grain size distribution. Thus, testing data of the limits should be added, and the overall grain size distribution of clay-silt mixtures will be better than the separated grain size distribution.

Answer: 4. Thanks very much for the opinions given by the reviewer. Many factors are not considered or taken seriously before. After measurement, the ranges of plastic limit and liquid limit are provided. The plastic limit and liquid limit values are definitely different under different clay contents. The range of plastic limit values for hydrate-bearing sediment samples with different viscosities is 21.4% -25.6%, the range of liquid limit values is 56.1% -60.3%, and the range of residual moisture content is about 12%.

Maybe the overall grain size distribution of clay-silt mixtures will be better than the separated grain size distribution. However, considering the different contents of clay minerals, mineral compositions, and particle size ratios, the overall grain size distribution of clay-silt mixtures is hard to exhibit here. The clay grain size distribution and the silt grain size distribution are shown separately. See lines 175-177 on Page 7.

Reviewer #2: The present paper intends to explore the mechanical properties of unconsolidated hydrate sediments, analyze variation laws and underlying reasons by considering hydrate saturation, effective confining pressure, clay content and clay type. This is helpful for wellbore instability analysis and sand production prediction, which is very interesting and meaningful. However, there are still few questions need to be clarified and discussed.

1. Common clay types include kaolinite, montmorillonite, illite and chlorite. This paper mainly discusses montmorillonite and illite, which are believed the main composition of shallow clay minerals in Shenhu sea area. What about kaolinite and chlorite? The type, content and distribution characteristics of clay minerals in Shenhu sea area should be supplemented detailedly in the introduction.

Answer: 1. Through literature review, the main components of hydrate-bearing sediments in the Shenhu Sea area are detrital minerals, clay minerals, and carbonate minerals. And, the main components of clay minerals are montmorillonite and illite, with a small amount of chlorite and kaolin. For example, the montmorillonite content in SH2B reservoir is 33% -59%, with an average of 47.04%; Illite is 22% -39%, with an average of 29.28%; Chlorite is 9% -17%, with an average of 13.17%; Kaolinite is 7% -14%, with an average of 10.51%. It can be seen that the content of chlorite and kaolinite is significantly different from that of montmorillonite and illite, so the impact of chlorite and kaolinite on the mechanical properties is not analyzed in the paper. See lines 45-46 on Page 2.

2. Some details of the experiment need to be discussed. Are the gas hydrate samples prepared and tested in one cell or in different? What is the initial water saturation? What methods are used to prepare uniform samples?

Answer: 2. The hydrate-bearing samples are prepared and tested in one cell. The initial water saturation can be calculated by Eq.(1). To prepare uniform hydrate-bearing samples, the following methods are used: In the proportionally prepared mixture of sand and clay, spray the corresponding mass of deionized water with a spray can, and keep stirring during this process until the water and material are evenly mixed. Then seal it with a plastic bag for 24 hours to evenly distribute the moisture in the material. Next, the material is filled into the mold, and the filling process is divided into five layers for compaction. After each layer is compacted, the surface is roughened with fine iron wire before continuing to fill, in order to prevent obvious layering interfaces between the two layers and affect the experimental results.

3. In the paper, there is no information on the place where the temperature is measured. In addition, temperature has a significant impact on the mechanical properties of hydrate sediment. The higher the clay content, the more obvious the effect may be. What is the temperature variation during the mechanical testing? Would this variation have an impact on test results and model establishment? The effect of temperature on unconsolidated hydrate sediments is not considered in this paper, and can be the future research direction (I share a good idea and conception with the Authors).

Answer: 3. Thanks for your suggestions. The information of the triaxial equipment is supplemented, and the temperature sensing devices are installed at the center of cold storage and inside the high-pressure chamber of triaxial equipment, the temperatures are measured. During the mechanical testing, the temperature will fluctuate and show a decreasing trend, but the ambient temperature is constant and the overall fluctuation range is limited. The changes in temperature during this process and the impact on the mechanical properties of hydrate-bearing sediment are complex. Therefore, the impact of temperature has not been considered in this paper, and further research may be conducted on this topic in the future.

4. In the experiment, how to get the desired hydrate saturation by hydrate depressurization decomposition? Does the hydrate samples remain uniform during the process?

Answer: 4. According to the photographic equilibrium curve, the hydrate in the sample is decomposed to different saturation levels by adjusting the effective confining pressure. When the hydrate no longer decomposes and the pressure stabilizes, the amount of gas decomposed is determined, and the remaining hydrate saturation is calculated based on the Van der Waals actual gas state equation.

During this depressurization process, the external part of the hydrate sample is first affected, then the impact is transmitted to the internal part. However, due to the small size and high porosity of the sample, the pressure transmission is rapid, and the degree of decomposition affected by the pressure is definitely more thorough, and the sample is relatively uniform.

5. This paper gives the influence of two clay minerals, montmorillonite and illite, on the mechanical properties and strength parameters, which clay mineral has greater influence? Why?

Answer: 5. From Fig. 8, when the content ratio of montmorillonite/illite decreases, the illite content increases, and the peak strength of hydrate-bearing sediment shows an increasing trend, indicating that the cementation of illite is stronger than that of montmorillonite. This is because the particle size of illite is bigger than that of montmorillonite, the specific surface area is smaller, leading to weaker water absorption and swelling properties, the bonding strength of illite crystal layer is greater. Moreover, the hydration film of illite mineral is thinner, and the friction resistance between particles is greater during sliding. See lines 239-248 on Page 10 and Page 11.

6. The latest references are listed in the paper. However, there are few format issues in the reference. Carefully proofread the reference and unify the author's name citation format.

Answer: 6. The references have been updated, the author’s name citation format has been unified. See lines 407-557 on Page 17 to Page 22.

---

## [Decision Letter · Decision Letter 1]

10 Feb 2025

Mechanical Properties Research of Unconsolidated Hydrate-Bearing Sediments under the Effect of Clay Minerals

PONE-D-24-35292R1

Dear Dr. Sun,

We’re pleased to inform you that your manuscript has been judged scientifically suitable for publication and will be formally accepted for publication once it meets all outstanding technical requirements.

Kind regards,

Jianguo Wang, PhD

Academic Editor

PLOS ONE

Additional Editor Comments (optional):

Reviewers' comments:

Reviewer's Responses to Questions

**Comments to the Author**

1. If the authors have adequately addressed your comments raised in a previous round of review and you feel that this manuscript is now acceptable for publication, you may indicate that here to bypass the “Comments to the Author” section, enter your conflict of interest statement in the “Confidential to Editor” section, and submit your "Accept" recommendation.

Reviewer #2: (No Response)

2. Is the manuscript technically sound, and do the data support the conclusions?

Reviewer #2: (No Response)

3. Has the statistical analysis been performed appropriately and rigorously? 

Reviewer #2: (No Response)

4. Have the authors made all data underlying the findings in their manuscript fully available?

Reviewer #2: (No Response)

5. Is the manuscript presented in an intelligible fashion and written in standard English?

Reviewer #2: (No Response)

6. Review Comments to the Author

Reviewer #2: (No Response)

7. PLOS authors have the option to publish the peer review history of their article (what does this mean? ). If published, this will include your full peer review and any attached files.

**Do you want your identity to be public for this peer review?** For information about this choice, including consent withdrawal, please see our Privacy Policy .

Reviewer #2: No

---

## [Editor Report · Acceptance letter]

PONE-D-24-35292R1

PLOS ONE

Dear Dr. Sun,

I'm pleased to inform you that your manuscript has been deemed suitable for publication in PLOS ONE. Congratulations! Your manuscript is now being handed over to our production team.

Kind regards,

on behalf of

Dr. Jianguo Wang

Academic Editor

PLOS ONE